# Deep Generative AI for Multi-Target Therapeutic Design: Toward Self-Improving Drug Discovery Framework

**DOI:** 10.3390/ijms262311443

**Published:** 2025-11-26

**Authors:** Soo Im Kang, Jae Hong Shin, Benjamin M. Wu, Hak Soo Choi

**Affiliations:** 1Institute for Cancer Genetics, Columbia University Irving Medical Research Center, 1130 St. Nicholas Ave, New York, NY 10032, USA; 2Qunova Computing Inc., Chrono Building Suite 501, 316 Gajeongro, Daejeon 34130, Republic of Korea; 3Craniofacial Biology and Bioengineering, ADA Forsyth Institute, 100 Chestnut St, Somerville, MA 02143, USA; 4Bioengineering and Nanomedicine Program, Department of Radiology, Massachusetts General Hospital and Harvard Medical School, 55 Fruit Street, Boston, MA 02114, USA; 5School of Materials Science, Japan Advanced Institute of Science and Technology, 1-1 Asahidai, Nomi 923-1292, Ishikawa, Japan

**Keywords:** deep generative model, autonomous drug discovery, polypharmacology, multi-target drug design, reinforcement learning, self-improving framework

## Abstract

Multi-target drug design represents a paradigm shift in tackling the complexity and heterogeneity of diseases such as cancer. Conventional single-target therapies frequently face limitations due to network redundancy, pathway compensation, and adaptive resistance mechanisms. In contrast, deep generative models, empowered by advanced artificial intelligence algorithms, provide scalable and versatile platforms for the *de novo* generation and optimization of small molecules with activity across multiple therapeutic targets. This review provides a comprehensive overview of the recent landscape of AI-driven deep generative modeling for multi-target drug discovery, highlighting breakthroughs in model architectures, molecular representations, and goal-directed optimization strategies. We also examine the emergence of self-improving learning systems, closed-loop frameworks that iteratively refine molecular candidates through integrated feedback, as a transformative approach to adaptive drug design. Finally, key challenges, current limitations, and emerging trends are discussed to guide the evolution of next-generation intelligent and autonomous drug discovery pipelines for multi-target therapeutics.

## 1. Introduction

The pharmaceutical industry is undergoing a groundbreaking change through the integration of artificial intelligence (AI) into drug discovery processes and healthcare sectors [1,2]. The successful application of AI techniques in drug discovery revolutionized AI-based approaches for *de novo* drug design, co-folding-based drug-target 3D structure prediction, drug-target interaction (DTI) modeling, and binding affinity estimation [2,3]. The integration of AI signifies a major paradigm shift, addressing the long-standing challenges of conventional drug discovery, complexity, high cost, and time-consuming process, often exceeding a decade and billions of dollars [4,5,6]. Conventional approaches typically involve extensive testing and sequential stages, imposing substantial resource and time burdens on pharmaceutical companies. Despite these substantial investments, fewer than 4% of new drugs intended for cancer treatment gain FDA approval annually, reflecting the inherent difficulty and limited mechanistic understanding of how drugs interact with the complex mechanisms of cancer [7,8].

AI-driven drug discovery, particularly employing deep generative models (DGMs), has become practical due to advances in computing power, big data analytics, and sophisticated algorithms. AI approaches, especially machine learning (ML) and deep learning (DL) algorithms, have emerged as powerful tools for processing vast multi-omics and chemical datasets, enabling the discovery of hidden biological patterns and identification of promising therapeutic candidates [2,3,4]. These technological advancements allow for more efficient, cost-effective, and innovative approaches to drug discovery and development [4,5,6]. ML and AI techniques now facilitate multi-target drug design by predicting molecules capable of modulating several proteins simultaneously, thereby improving efficacy, reducing off-target effects, and minimizing adverse events [9,10,11,12]. The integration of multi-omics, structural, and patient-specific data further enhances predictive accuracy and advances the field of precision medicine.

Iterative AI frameworks continually optimize molecular structures to improve efficacy, pharmacokinetics, and safety profiles, with the ultimate aim of achieving optimal pharmacological outcomes. The concept of multi-target generative design arises from the observation that single-target therapeutics, although initially effective, often lose efficacy due to compensatory signaling pathways, adaptive resistance, and biological redundancies [7,10,11,12]. Complex diseases such as cancer involve interdependent molecular networks, where targeting a single node frequently fails to achieve durable therapeutic effects. Generative models capable of constructing compounds with polypharmacological activity offer an opportunity to address these challenges by exploring high-dimensional chemical spaces and generating molecules that balance potency, selectivity, and safety across multiple targets. This multi-target strategy aligns with the systems-level understanding of disease biology, promoting network stabilization rather than reliance on single pathway inhibition.

The development of multi-target therapeutics, however, remains substantially more complex than single-target optimization. Recent breakthroughs in AI, particularly in reinforcement learning (RL), have opened new avenues in multi-target therapeutic design [3,4]. RL introduces a dynamic decision-making process into molecular generation, allowing models to iteratively maximize reward functions that integrate multiple pharmacological objectives, such as target affinity, drug-likeness, and toxicity minimization. In multi-target drug design, RL enables adaptive optimization across conflicting objectives and balances trade-offs through weighted composite rewards. This capacity to learn from feedback empowers models to autonomously navigate vast chemical spaces and identify molecules with optimized multi-target profiles. Moreover, coupling RL with active learning (AL) within a Design-Make-Test-Learn (DMTL) cycle fosters continuous co-improvement of generative and predictive systems, accelerating convergence toward high-quality lead discovery.

This review explores recent advances in AI-driven, self-improving frameworks for multi-target drug design, emphasizing the synergistic roles of deep generative modeling and RL. A systematic literature search was conducted using PubMed, Scopus, Web of Science, and Google Scholar for studies published between 2010 and 2025, focusing on the keywords: AI-driven drug discovery, deep generative models, reinforcement learning, multi-target therapeutics, and self-improving AI. These self-optimizing architectures integrate data-driven generation, feedback learning, and predictive adaptation to enhance molecular novelty, potency, and safety. Remarkably, self-improving AI systems equipped with RL and AL have demonstrated the ability to increase chemical diversity, improve predictive robustness, and autonomously optimize candidate molecules. Importantly, these advanced methods have broad applicability beyond oncology, extending to other diseases characterized by complex network-level dysregulation.

As illustrated in Figure 1, the self-improving workflow demonstrates a closed-loop framework combining DGMs, RL, and AL to efficiently generate, evaluate, and refine multi-target candidates. RL functions as the adaptive core, guiding molecular exploration toward desired pharmacological profiles, while AL continuously updates predictive models through uncertainty-driven sampling, establishing a dynamic and intelligent framework for autonomous multi-target drug discovery.

## 2. Evolution of Drug Discovery Paradigms

Drug discovery aims to design novel molecules with specific physicochemical and biological properties suitable for clinical evaluations [13]. However, over the past decades, it has remained resource-intensive and time-consuming, significantly limiting the systematic exploration of chemical space [13,14,15]. Advances in the understanding of complex disease networks have further underscored the limitations of the traditional one-drug-one-target paradigm. As a result, single-target drug discovery is increasingly shifting toward multi-target strategies designed to modulate various biological pathways [10,16]. This transition reflects a fundamental change in approach, acknowledging that many diseases are driven by complex, interconnected molecular networks [7,16]. Within this evolving framework, the ability to identify interactions between drugs and protein targets remains essential for both drug design and drug repurposing [7,16]. Due to experimental approaches for mapping such interactions often being labor-intensive, costly, and low-throughput, in silico prediction has become a practical and complementary strategy for prioritizing candidates and guiding experimental efforts [7,10,16]. Advances in AI, particularly deep generative modeling, now offer promising strategies for exploring chemical space and designing multi-target drugs more efficiently.

## 3. Deep Generative Models in Molecular Design

### 3.1. Molecular Representations

Robust molecular representations are indispensable for the success of generative models and for ensuring interpretability by chemists. The Simplified Molecular-Input Line Entry System (SMILES) remains a popular method for encoding molecular structures in a compact, partially human-interpretable string format. Most generative models utilize SMILES for both training and generating novel molecules, particularly in de novo drug design applications [3,4]. In contrast, molecular graphs represent molecules by mapping atoms as nodes and bonds as edges, enabling efficient computational operations and facilitating tasks such as substructure searching and topological descriptor calculations. However, molecular graphs lack the 3D spatial information necessary for accurately modeling stereochemistry and describing ligand–protein interactions within binding sites. The geometric arrangement of atoms, including bond and torsion angles, provides essential steric and spatial context for these structure-based applications.

Unlike 2D graphs, 3D molecular representations incorporate geometric and steric data, often leveraging equivariant neural networks or diffusion models trained on conformer ensembles to produce physically meaningful embeddings (Figure 2A). These methods are critical for tasks demanding spatial precision, such as molecular docking and structure-based drug design [3,4,17]. Within this landscape, the SELFIES representation offers a distinctive advantage. SELFIES is a fully robust molecular string format in which every sequence corresponds to a valid chemical structure, making it particularly attractive for generative modeling and data-driven inverse design. Its guaranteed validity enables seamless integration with ML pipelines and ensures consistent translation to 3D molecular structures for downstream analysis. By comparison, SMILES is syntactically fragile and often generates invalid strings [18].

Beyond molecular representation, integrating RL and AL within the DMTL framework fosters the emergence of self-improving molecular design systems. As shown in Figure 2B, the generative model first produces novel molecular candidates, guided by conditioning vectors such as desired target profiles or activity constraints, to ensure goal-directed output. These candidate molecules then progress to the Test phase, where they are evaluated using predictive models (acting as *in silico* oracles) that perform structure-based simulations, scoring properties like activity, toxicity, and drug-likeness. The results generate a reward signal used by RL to refine the generative model’s strategies, maximizing multi-objective reward functions and optimizing the generation policy. Concurrently, AL selects the most informative compounds, those with high predictive uncertainty or structural novelty, for further data acquisition, such as experimental testing or high-fidelity simulations. The resulting data is fed back into model retraining, accelerating the DMTL cycle. By enabling co-optimization of both generative (RL-guided) and predictive (AL-driven) components, it dramatically improves the efficiency and success rate of multi-target lead discovery.

### 3.2. Deep Generative Model Architectures

DGMs have grown significantly in popularity due to their ability to learn and mimic complex data distributions [9,19]. These provide powerful computational tools for exploring vast chemical space, new for molecular discovery [20]. In the past few years, *de novo* molecular design has increasingly leveraged DGMs from the emergent field of DL, proposing novel compounds with a high likelihood of possessing desired pharmacological properties and activities [17]. A variety of DGM architectures, including Recurrent Neural Networks (RNNs), Variational Autoencoders (VAEs), Generative Adversarial Networks (GANs), transformers, and diffusion models, can learn from existing datasets to facilitate the generation of novel compounds [17]. RNNs are widely employed in generative models to process sequential, string-based representations. In molecular design, this involves learning structural patterns from SMILES strings to generate chemically valid and diverse molecules, providing a natural and effective framework for sequence-based molecular generation. Variants such as Long Short-Term Memory (LSTM) and Gated Recurrent Unit (GRU) networks are particularly effective due to their ability to capture long-range dependencies within molecular sequences [3,4]. VAEs learn a probabilistic latent space representation of chemical structures, allowing for both the reconstruction of training molecules and the generation of novel structures by encoding to and decoding from the latent vectors [4]. GANs, while widely applied in image synthesis and scaled up for large-scale applications such as language modeling, notably the Optimus VAE language model [21], have been adapted for molecular generation, where a discriminator guides the generator toward producing realistic chemical structures.

Transformer-based models, typified by architectures such as GPT, have demonstrated strong performance in molecular design tasks using SMILES representations. By leveraging self-attention mechanisms instead of recurrent structures, these models effectively capture long-range dependencies in molecular sequences and have found recent applications in *de novo* generation, scaffold-constrained design, and multi-objective molecular optimization. They excel in expressive chemical language modeling, while diffusion models offer robust capabilities for 3D structure generation [22]. Diffusion models, a class of probabilistic generative algorithms, are rapidly gaining prominence in both image and molecular generation. These models learn to reverse a gradual noising process applied to data, enabling the production of highly realistic samples and presenting unique advantages over traditional models such as GANs and VAEs in drug discovery [3,4,5,6,17]. Notably, diffusion models excel at generating diverse molecular structures, especially in three dimensions, through equivariant geometric modeling and strong conditional control, including the imposition of protein pocket, pharmacophore, or property constraints [23]. Nevertheless, challenges persist, including the assurance of structural validity, targeted biological activity, and synthetic feasibility of generated molecules. Furthermore, the computational demands of diffusion models can be substantial, arising from the numerous iterative steps required for both forward (diffusion) and reverse sampling processes [19]. There is a marked shift in the scientific community toward transformer and diffusion models due to their superior scalability, training stability, and capacity to model complex long-range molecular interactions, which leads to more accurate and controllable molecular generation.

### 3.3. Comparative Analysis of Deep Generative Model Architectures

The generative model architectures described above differ in their respective strengths and limitations, shaping their applicability to multi-target drug design. RNNs offer straightforward implementation and high interpretability but are hampered by poor scalability, sequential bottlenecks, and susceptibility to error propagation. VAEs benefit from a continuous latent space well-suited to property optimization, though weak regularization can reduce molecular validity or diversity. Transformers outperform earlier architectures due to their scalability and ability to model long-range chemical relationships, making them suitable for large datasets and complex multi-objective tasks; however, they incur greater computational costs and somewhat reduced interpretability. Diffusion models achieve remarkable validity, diversity, and realistic 3D generative capabilities via iterative denoising, yet require extensive computational resources and high-quality training data [3,4]. Performance across all architectures is profoundly influenced by data quality and benchmark consistency. Standardized datasets such as MOSES, GuacaMol, and ZINC support reproducibility, but real-world chemical datasets frequently present noise, imbalance, and annotation discrepancies. Further challenges, such as hyperparameter tuning, multi-objective reward balancing, and integration of structural representations, emphasize the need for improved data curation and standardized evaluation protocols. These will ensure the robustness and translational impact of AI-driven molecular generation.

### 3.4. Goal-Directed Generation and Guided Optimization

Conditioning and control strategies within generative AI models are essential for producing chemically valid and pharmacologically relevant molecules, significantly reducing the need for complex post hoc filtering and facilitating seamless end-to-end learning [10,11,12]. This is especially crucial in the context of multi-objective optimization, which demands satisfaction of multiple pharmacological criteria simultaneously. These mechanisms are powered by three main inputs: (1) target embeddings (encoded protein features), (2) activity profile vectors (representing multi-target bioactivities), and (3) scaffold constraints (retaining molecular cores during optimization for selectivity and ADMET).

Conditioning vectors can be sourced from diverse datasets according to biological, chemical, and model-specific needs. These vectors may be derived from experimental or curated datasets (such as binding affinities, inhibition constants, or activity annotations), engineered molecular descriptors (fingerprints, physicochemical properties, or family-specific feature sets), learned embeddings from pretrained protein language models, structure-based encoders, or graph neural networks that capture high-dimensional biological and structural relationships. Target embeddings provide compact, informative representations of protein features, summarizing structural and functional attributes essential for ligand design [24].

In practice, conditioning signals can be integrated through three technical mechanisms. (1) Latent vector concatenation, appending activity or property endpoints to the latent code in VAE or flow-based models to impose target-specific constraints. (2) Cross-attention conditioning, enabling transformers to incorporate target embeddings or property vectors as key–value pairs, thereby modulating token- or graph-level generation. (3) Diffusion-based conditioning, using conditioning vectors to modulate the denoising trajectory via classifier/classifier-free guidance, conditional score networks, or timestep-wise feature modulation. These mechanisms support precise and controllable multi-target or property-driven molecular generation. For example, transformer-based multi-target generative models extend these concepts by employing latent protein representations, such as those derived from AlphaFold3 or other protein language/structure models, to condition ligand generation on broad or precisely defined target ensembles [25]. Such embeddings may be extracted from amino acid sequences or engineered from structural motifs, allowing DGMs to target various protein classes (e.g., serine/threonine or tyrosine kinases), broadening the diversity of generated molecules [23,24,25]. End-to-end models utilizing protein sequences alone further demonstrate the feasibility of generating novel, bioactive compounds without the need for detailed structural data. Strategies for embedding protein features thus significantly expand the versatility and biological relevance of DGMs in multi-target drug design [26,27,28].

The challenge in multi-target design extends beyond target identification to a comprehensive understanding of desired bioactivity patterns, addressed by activity profile vectors. These vectors quantitatively or qualitatively represent compound activity across panels of biological targets, including functional activity parameters such as IC_50_ values or binary labels, collectively forming a compound’s pharmacological fingerprint. Used as conditional inputs, they guide the generative process towards molecules with tailored bioactivity spectra. When combined with target feature embeddings, both molecular context (e.g., scaffold or substructure constraints) and biological outcomes (e.g., multi-target efficacy) can be simultaneously specified [29,30]. When combined with target feature embeddings, this allows for simultaneous specification of both the molecular context (molecular scaffold or substructure to include) and the biological outcome (bioactivity to achieve across the target panel) [25].

Scaffold constraints are particularly important for medicinal chemistry, preserving core molecular frameworks while facilitating systematic exploration of substitutions and functional group modifications [31]. Graph-based generative models can elaborate molecules from predefined scaffolds, systematically probing feasible chemical elaborations and maintaining the central core [32,33]. These constraints empower medicinal chemists to optimize selectivity, off-target effects, and drug-likeness, promoting lead generation within established chemical series. They can be incorporated in RL frameworks via penalty functions that deter deviation from the scaffold or filtering steps that retain core structures during optimization [31,32,33,34].

Nonetheless, excessive constraint may cause mode collapse, where model outputs lose diversity and novelty, limiting exploration of chemical space and potentially biasing towards known chemotypes. Solutions such as diversity-increasing rewards, latent-space regularization, and scaffold-hopping modules are being developed to address these issues while preserving clinical relevance. Collectively, these conditioning mechanisms, target embeddings, activity profile vectors, and scaffold constraints, create an integrated framework for precise, multi-objective molecular design, enabling generative models to produce therapeutically relevant, synthetically feasible compounds aligned with the complex requirements of multi-target drug discovery [31,32].

## 4. Self-Improving Systems in AI-Driven Drug Design

### 4.1. The Principle of Self-Improvement

The swift progress of AI has enabled the development of self-improving algorithms that allow AI systems to enhance their capabilities autonomously once initiated [35]. These algorithms are essential for empowering AI systems to gather knowledge and refine their operations by analyzing real-time data and updating models with new insights. This approach is highly valuable in drug discovery, where continuous learning from experimental outcomes is crucial for improving model performance. Specifically, self-improving algorithms learn from the outcomes of experimental validation, incorporating feedback on factors such as synthetic feasibility, bioactivity, and other properties to continuously refine future predictions [35]. This creates a virtuous cycle where experimental results directly inform computational models, leading to the generation of better candidates for the next round of experimental testing.

### 4.2. The Design-Make-Test-Learn (DMTL) Loop

Self-improving systems operationalize the DMTL loop, a cyclical, data-driven framework facilitating the rapid optimization of complex molecular systems through iterative learning and feedback. By integrating AI-driven modeling, automated synthesis, and high-throughput screening with multi-objective decision-making, this loop streamlines lead optimization across pharmacological, structural, and physicochemical properties, thus accelerating modern drug discovery efforts, encompassing the following four steps: (1) Design: DGMs, such as VAEs, GANs, and transformers, propose novel molecules based on multi-target profiles, ADMET constraints, and domain-specific priors. (2) Make: This phase may leverage robotic synthesis platforms or rigorous *in silico* filtering (e.g., retrosynthesis planning, synthetic accessibility scoring) to shortlist feasible candidates. (3) Test: Compounds are either experimentally validated or scored using surrogate models such as molecular docking, quantitative structure–activity relationship (QSAR) models, binding free energy calculations, and physicochemical property predictors. (4) Learn: The test outcomes are fed back to refine the generative and predictive models using supervised fine-tuning, RL, or AL strategies. This self-improving feedback loop allows for continuous and efficient exploration of promising chemical spaces while minimizing experimental costs. When embedded into a multi-objective, multi-target framework, the DMTL loop provides a dynamic mechanism for balancing efficacy, selectivity, safety, and novelty, offering a significant advantage in drug development [35,36]. Figure 3 illustrates a closed-loop AL strategy integrated with DGMs for the rational design of multi-target inhibitors. The process is initiated with a labeled molecule (e.g., binding affinity), starting the DMTL cycle to generate novel chemical structures. Candidates undergo experimental testing, and informative data points are intelligently queried to retrain the generative model. Through this iterative refinement, molecules are optimized for improved polypharmacological profiles, enhancing binding to key targets and minimizing off-target toxicity. The overall process enables data-efficient exploration of chemical space, accelerating the development of selective and potent inhibitors against the target networks implicated in complex diseases, including cancer.

### 4.3. Self-Improving Algorithms for Optimizing Drug Design

The conditioning (protein features and activity profiles) provides directional guidance for generating molecules. It aligns with complex, multi-objective requirements, including activity, selectivity, and toxicity, which are critical for multi-target optimization. Conditioning thus plays a central role in self-improving drug design frameworks by offering an interpretable mechanism to control molecular generation, but the iterative refinement of the generative process itself relies on specialized self-improving algorithms: RL and AL. RL trains generative models using a reward function that guides them toward generating molecules with optimal properties. In this framework, the generative model is treated with a policy $\pi_{\theta }\left(a_{t}|s_{t}\right)$, where $s_{t}$ is the current molecular state and $a_{t}$ is the action (e.g., addition of a new molecular fragment). The generative model receives a reward $r_{t}$, and aims to maximize the expected return:$$\;\;J\left(\theta \right)=E_{\pi_{\theta }}\left[\sum_{t=0}^{T}r_{t}\right]$$

This is typically optimized using the REINFORCE algorithm:$$\nabla_{\theta }J\left(\theta \right)=E_{\pi_{\theta }}\left[\sum_{t=0}^{T}\nabla_{\theta }log\pi_{\theta }\left(a_{t}|s_{t}\right)G_{t}\right]$$
where $G_{t}=\sum_{k=0}^{T-t}\gamma^{k}r_{t+k}$ is the discounted cumulative reward and γ ∈ [0, 1] is the discount factor.

In drug discovery applications, the reward $r_{t}$ is typically composed of a weighted combination of molecular properties:$$r_{t}\;=\;\lambda_{1}\;\mathrm{QED}\;+\;\lambda_{2}\;{{\mathrm{pIC}}_{50}}^{\mathrm{(target)}}\;-\;\lambda_{3}\;\mathrm{hERG}\;\mathrm{risk}\;-\;\lambda_{4}\;\mathrm{SAS}\;+\;\lambda_{5}\;\mathrm{Novelty}$$where $\lambda_{i}$ are hyperparameters that weight the importance of various objectives such as the quantitative estimate of drug-likeness (QED), synthetic accessibility score (SAS), cardio toxicity (e.g., hERG), and structural novelty [37,38,39]. Multi-objective RL frameworks facilitate the simultaneous optimization of multiple endpoints, including maximized efficacy while minimizing the toxicity and assurance of synthetic feasibility. Toxicological liabilities, such as hERG inhibition, are incorporated directly into the reward function by penalizing candidate molecules predicted to be unsafe [40].

Scaffold-constrained generative models often experience mode collapse and limited novelty because fixing the core structure restricts the generative space and drives the model to repeatedly produce a small set of high-probability R-group patterns from the training data, resulting in minimal structural innovation. To counter this, structural novelty and scaffold diversity can be enhanced by incorporating similarity penalties, such as Tanimoto-based constraints or scaffold-level filters, which discourage redundant generations and help mitigate mode collapse. To account for synthetic accessibility, SAS are integrated into the reward formulation to penalize structurally complex or infeasible compounds [41,42]. Certain workflows further employ DL-based predictors to evaluate synthetic tractability, while others complement post hoc filtering or retrosynthetic analysis. By integrating these components into a unified, multi-objective reward scheme, RL-guided generative models are capable of producing candidate molecules that are novel, structurally diverse, and synthetically viable [38].

AL and uncertainty-aware sampling strategies play a pivotal role in data-efficient drug discovery, especially under conditions of data sparsity and high experimental cost [39]. By prioritizing candidates that maximize information gain, these approaches enable more efficient utilization and enhance the robustness of predictive models. The goal is to identify molecules that are expected to yield the greatest gain in predictive performance if labeled and added to the training set. Let $D_{L}$ and $D_{U}$ denote the labeled and unlabeled datasets, respectively. Given a predictive model f^ (x), molecules are selected according to an acquisition function. For uncertainty sampling, molecules with the highest predictive uncertainty σ(x) are selected:$$x^{*}\;=argmax_{x\in D_{U}}\sigma \left(x\right)$$

In Bayesian frameworks (e.g., deep ensembles, MC Dropout), predictive uncertainty is estimated as:$$\sigma^{2}\;(x)\;=\;E\;[f\;(x{)}^{2}]\;-\;{\left(E\;[f\;(x)])\right]}^{2}$$

To ensure chemical diversity, AL frameworks also incorporate a novelty term:$$x^{*}\;=\;argmax_{x\in D_{U}}min_{x^{\prime }\in D_{L}}Sim\left(x,\;x^{\prime }\right)$$, where Sim is a molecular similarity metric, commonly computed as the Tanimoto coefficient over ECFP4 fingerprints. These two acquisition objectives, uncertainty and novelty, can be combined using a multi-objective acquisition function:$$x^{*}\;=\;argmax_{x\in D_{U}}\;[\alpha \cdot \sigma \;(x)\;+\;(1\;-\;\alpha )\cdot \mathrm{Novelty}\;(x)]$$

This balanced strategy enables AL to explore underrepresented regions of chemical space, improving model generalization across diverse chemical scaffolds and ensuring robust multi-target performance [43]. In practice, contemporary AL systems often balance uncertainty and novelty, either sequentially or through multi-objective acquisition functions, ensuring both information gain and chemical diversity in each iteration [39,44]. By methodically augmenting the training set with the most informative data points, AL accelerates the DMTL cycle and is also designed to prioritize molecules that feature novel structural scaffolds, moving beyond minor analogs or variations on previously encountered chemotypes. AL and uncertainty-aware exploration offer a principled alternative by systematically prioritizing molecules with high predictive uncertainty or novel scaffolds. These strategies not only address data sparsity but also enhance model generalizability, accelerate learning, and improve the translational relevance of predictive models. Through iterative refinement and adaptive sampling, AL is emerging as a foundational approach in molecular discovery and design, enabling efficient navigation of chemical space and facilitating complex, multi-target optimization tasks in AI-driven discovery.

As depicted in Figure 4, AL is increasingly integrated with omics-based profiling, pharmacological network analysis, and biological pathway modeling to guide rational molecular design in complex diseases such as cancer. Within this framework, the generative model is conditioned on disease-specific biological signatures, derived from genomic, transcriptomic, and proteomic data, to design molecules with desired activity across multiple oncogenic targets. Informative compounds identified through AL are synthesized or simulated for testing, and the feedback is incorporated to retrain the generative model. This closed-loop, uncertainty-driven refinement process enables efficient exploration of chemical space and supports the rational design of polypharmacological agents targeting interconnected oncogenic networks, such as those involving PI3K, BRD4, and related signaling pathways. Ultimately, AL facilitates dynamic knowledge integration across data modalities, enabling AI-driven, multi-target drug discovery.

## 5. Applications and Case Studies

The growing understanding of complex diseases has driven drug discovery from a one-target one-drug model to a multi-target multi-drug model for simultaneous modulation of multi-targets. This is particularly critical in contexts like cancer, where combination therapy is essential for improving overall response rates [10,12,20,26]. However, as the number of therapeutic options and target pairs increases, identifying an optimal drug combination becomes exponentially infeasible using conventional experimental methods. This search complexity, compounded by patient heterogeneity, underscores the necessity for computational approaches [20,26,39]. Polypharmacology, the strategy of modulating multiple targets with a single agent (multi-targeted monotherapy), has thus emerged as a vital paradigm in modern drug discovery, offering enhanced efficacy and a potentially improved safety compared to single-targeted approaches [44].

### 5.1. Applications in Multi-Target Cancer Therapeutics

**Kinase Inhibitor Design:** Protein kinases are central regulators of cellular processes like proliferation and survival are frequently dysregulated in cancer [45,46]. DGMs have become powerful tools for the intelligent design of multi-target kinase inhibitors with precisely defined affinity profiles [46,47]. A notable application is the design of dual inhibitors for CDK4/6 and CDK9 [48]. CDK4/6 inhibition blocks cell cycle progression, while inhibition of CDK9 suppresses the transcriptional elongation essential for tumor cell survival and drug resistance. Novel compounds designed by generative models achieve this dual inhibition, combining antiproliferative effects with the suppression of critical transcriptional programs for a synergistic, robust block on cancer cell survival and cancer adaptation [48,49]. Other successful examples include the use of Perturbation Theory combined with ML (PTML) to design virtual dual inhibitors for CDK4 and HER2 [49]. Similarly, the development of PI3K/mTOR dual inhibitors targets a pathway known for redundancy and feedback-driven resistance. DL-guided molecular generation prevents compensatory pathway reactivation, achieving deeper suppression of oncogenic signaling and translating to enhanced antitumor efficacy [15,50]. Furthermore, the established success of drugs like sunitinib, which inhibits various receptor tyrosine kinases (RTKs), including all platelet-derived growth factors (PDGFRs), highlights the importance of multi-kinase inhibitors as targets for DGM drug design [51]. Clinical evidence consistently supports that the strategy of polypharmacology leads to improved patient outcomes and delayed development of therapeutic resistance compared to single-target approaches [50,51]. In addition, Gackowski et al. demonstrated that a multivariate adaptive regression splines (MARSplines) QSAR model can identify a polypharmacological target of Stevioside, a natural sweetener, including its inhibitory activity against activated coagulation factor X (FXa) [52].

**Chemogenomics-Driven Design:** This design strategy advances the paradigm for multi-target therapeutics by integrating large-scale compound activity and structural data with multi-omics patient or tumor profiles (e.g., transcriptomic signatures and mutational landscape). This integration moves beyond a “one-size-fits-all” model, enabling patient-specific compound design tailored to the unique molecular features of individual tumors [53,54]. These workflows use ML to predict drug response based on features extracted from gene expression, mutation status, copy number alterations, and epigenetic modifications. By aligning molecular design with network pharmacology principles, chemogenomic pipelines identify patient-specific vulnerabilities and match them with candidate inhibitors that possess the desired multi-target activity spectrum [55]. Specifically, DGMs can be conditioned on patient- or subtype-specific omics signatures to propose novel compounds predicted to disrupt oncogenic networks at multiple, personalized levels.

**3D Printing and Personalized Multi-Target Therapeutics:** The convergence of 3D printing technologies and AI-driven drug design is speeding up the development of precision, patient-specific treatments [56,57]. 3D Pharming, defined as directly 3D printing pharmaceutical tablets, offers unprecedented possibilities for on-demand production of personalized medications [57]. Early clinical applications, such as the FDA-approved 3D printed tablet Spritam, demonstrate regulatory feasibility and real-world benefits. The combination of AI with 3D Pharming platforms promises to improve the flexibility, speed, and effectiveness of multi-target therapeutic design, representing a vital frontier for the next era of personalized medicine [58].

### 5.2. Benchmarking and Validation

Robust benchmarking and validation are essential to translate the output of generative models into practical therapeutic benefits. This relies heavily on large-scale public and proprietary databases.

**Compound Activity**: ChEMBL provides an extensive repository of small molecule structures and their measured activities across a wide array of molecular targets, facilitating the development and evaluation of AI-driven design workflows [56]. BindingDB complements this by supplying experimentally determined binding affinities, supporting the precise modeling of compound-target interaction patterns and cross-target selectivity.

**Genomics and Efficacy:** The Genomics of Drug Sensitivity in Cancer (GDSC) database contributes omics-anchored drug response information, enabling the prediction of compound efficacy across genetically diverse contexts and serving as a crucial benchmark for patient-specific design models.

**Synergy Tools:** Specialized tools are integrated for validating the synergy of multi-target agents. DrugComb compiles synergy screening data across cancer cell lines, supporting the systematic exploration of synergistic relationships. DeepSynergy employs a framework to predict synergistic drug responses by combining chemical structure and genomic features of cancer cell lines. Performance benchmarking using these tools, comparing predicted synergy scores with experimentally measured ones, is vital for validating the real-world polypharmacological potential of generated compounds [58,59].

### 5.3. Modern Approaches for Structure-Based Multi-Target Drug Design

A major focus in recent generative modeling research has been the development of methods that can jointly optimize multiple design objectives relevant to structure-based drug discovery.

**Structure-based Generative Modeling:** IDOLpro addresses a key limitation of existing generative models, the challenge of simultaneously meeting diverse pharmacological and physicochemical requirements. By integrating a diffusion-based generative process with multi-objective optimization, IDOLpro uses differentiable scoring functions to steer latent variables toward optimal regions of chemical space. This allows for efficient exploration of underrepresented molecular regions while jointly optimizing for binding affinity, drug-likeness, and synthetic accessibility [60]. In parallel, dedicated multi-target generative models are emerging to design compounds with activity across multiple biological targets. POLYGON [26], for example, performs latent-space optimization, jointly balancing predicted multi-target activity with synthetic feasibility. Other approaches, such as Chemical Language Models (CLMs) [61], use fine-tuning and pharmacophore fusion strategies to generate ligands from dual-target ligand sets. These ligand-centric approaches are particularly effective in polypharmacology, offering flexible frameworks to modulate multiple targets while maintaining favorable ADMET properties. Complementing these methods, structure-aware generative models incorporate 3D protein pocket information to guide ligand generation in a spatially precise manner. Pocket2Mol [62] and DiffSBDD [63] use point clouds or 3D spatial representations of binding pockets to generate ligands that conform to the local geometry. They leverage transformer-based or SE(3)-equivariant diffusion networks to ensure spatial fidelity.

**End-to-End Complex Prediction:** In tandem with ligand generation, accurate protein–ligand complex prediction plays a critical role in structure-based drug discovery. These models go beyond merely identifying binding pockets; they reveal key interacting residues, supporting mechanistic interpretation and enabling rapid Structure–Activity Relationship (SAR) development. End-to-end systems such as AlphaFold3 [25] and RoseTTAFold All-Atom (RFAA) [64] generate high-resolution 3D protein–ligand complexes directly from sequence or partial structural input. By integrating ligand-aware refinement, these models serve as foundational tools that support downstream generative workflows, significantly improving structural realism and design accuracy (Table 1). By integrating protein structure prediction, pocket-aware ligand generation, and multi-objective optimization, these models constitute a modern, versatile toolkit that advances the frontier of deep generative modeling in rational multi-target drug discovery.

### 5.4. Approaches to Ligand-Based Multi-Target Drug Discovery

A complementary strategy to structure-based design is the ligand-based approach. One example combines Perturbation Theory Machine Learning with an Ensemble of Multilayer Perceptrons (PTML-EL-MLP) [65] with the Fragment-Based Topological Design (FBTD) approach to predict triple-target inhibitors for cancer-related proteins. This method utilizes chemical and biological data from ChEMBL and employs the Box-Jenkins approach to generate multi-label topological indices, which train the PTML-EL-MLP model. The model achieved an accuracy of approximately 80%, enabling a chemistry-driven analysis of molecular fragments that positively influence multi-target activity. These favorable fragments were then used as building blocks to design new drug-like molecules, which were predicted to be triple-target inhibitors. This demonstrates the potential of combining PTML modeling with FBTD for generating chemical diversity in multi-target drug discovery in cancer research and beyond.

Pharmacophore modeling has also proven effective for multitarget discovery. Mendes et al. applied hierarchical pharmacophore-based screening to identify compounds with simultaneous AChE, BChE, and BACE-1 inhibition, underscoring the value of pharmacophoric feature patterns in polypharmacological ligand discovery [66]. Building on this idea of feature-level conditioning, Pharmacophore-Guided DL approach for bioactive Molecule Generation (PGMG) offers a modern extension for controllable molecule generation [67]. PGMG exemplifies this direction by conditioning generation on pharmacophore hypotheses encoded as complete graphs, using Gated GCNs to embed spatial feature constraints and a transformer decoder modulated by latent variables z to generate diverse molecules consistent with the conditional distribution p(x|p,z). Through RDKit-derived chemical features, shortest-path pharmacophore graphs, and infilling-corrupted SMILES, PGMG learns robust conditional mappings without target-specific labels and produces pharmacophore-aligned molecules with strong docking and ADMET performance across ligand-based, structure-based, and scaffold-hopping tasks.

In parallel, multitask learning (MTL) has enabled protein-conditioned ligand generation. DeepDTAGen integrates ligand–receptor interaction embeddings into a shared latent space jointly optimized for drug–target affinity (DTA) prediction and target-aware molecular generation [68]. Its gradient-alignment algorithm, FetterGrad, mitigates inter-task gradient interference and preserves heterogeneous and uniform affinity landscapes across diverse protein targets (e.g., Q9HAZ1, KIT, P61964; PAK4_HUMAN, RAF1, BRD4). These results highlight the capacity of DeepDTAGen to model target selectivity and generate biologically informed molecules tailored to specific protein interaction profiles.

Together, ligand-based approaches, including PTML–FBTD fragment design, pharmacophore-guided generative modeling, and MTL-based protein-conditioned generation, demonstrate that feature-level knowledge from ligands, pharmacophores, and ligand–receptor interactions can be harnessed to enable precise, constraint-driven multi-target and target-aware drug design.

## 6. Challenges and Limitations in AI-Driven Drug Design

Despite its transformative potential, the full integration of AI into practical drug discovery workflows is hindered by several critical challenges. A major obstacle is the limited interpretability of many high-performing models, particularly the deep neural networks used for prediction or molecular generation. Medicinal and computational chemists require mechanistic insight to validate model outputs and formulate new hypotheses, yet most current architectures function as opaque “black boxes,” offering limited explanatory value [3,40]. This lack of transparency complicates scientific understanding and undermines confidence in early-phase decisions. Although emerging efforts in Explainable AI (XAI), including attention mechanisms and counterfactual analysis, show promise, they remain underutilized in drug discovery.

Compounding the issue are persistent data quality and scarcity concerns. Public and proprietary datasets frequently contain noisy or sparse bioactivity labels, inconsistent assay protocols, and target annotation heterogeneity, all of which compromise model robustness and impair generalizability in multi-target settings [16,69]. Most public datasets (e.g., ChEMBL, BindingDB) are heavily biased toward well-studied targets, limiting the data available for novel or multi-target efforts. Structure-informed models face additional constraints owing to the scarcity of experimentally resolved protein–ligand complexes across diverse target classes.

Another practical concern is the synthetic feasibility of generated compounds. Many AI-generated molecules, while chemically valid, are synthetically infeasible due to unstable intermediates, overly complex substructures, or low-yield routes. This disconnect arises because most generative models prioritize idealized property optimization over real-world tractability. To address this, retrosynthesis tools such as ASKCOS (v3.0) [70] and AiZynthFinder (v4.4.0) [71], and reaction-based generation strategies, are being integrated into modern pipelines to enforce synthesis-aware design.

Technical and infrastructural barriers also limit deployment. State-of-the-art models, particularly those incorporating 3D spatial reasoning, diffusion processes, or end-to-end structure prediction, demand substantial computational resources. Furthermore, integrating AI tools into legacy pharmaceutical systems remains non-trivial due to incompatible data standards and a lack of interoperability [38,40,41]. Even with promising in silico results, wet lab validation remains a costly and time-consuming bottleneck, prone to failure due to unexpected toxicity or poor ADMET profiles. Current generative models primarily operate on 2D molecular graphs (SMILES, SELFIES, etc.), often overlooking the important 3D conformational diversity [3,4]. This diversity is crucial for target binding affinity, docking constraints, shape complementarity, and steric effects.

## 7. Conclusions and Future Prospects

Generative AI is rapidly transforming the landscape of therapeutic discovery. AI-driven drug design using DGMs for multi-target applications marks a significant breakthrough in pharmaceutical sciences. When integrated into self-improving, closed-loop frameworks, these technologies present robust, data-driven methodologies for molecular exploration, optimization, and validation of candidate therapeutics. Such AI-based approaches offer the potential to accelerate the identification of efficacious and safe drug candidates, dramatically reducing time and cost compared to traditional drug discovery pipelines.

The convergence of AI and drug design has ushered in a profound paradigm shift toward intelligent, learning-based systems capable of generating and refining therapeutic hypotheses across multiple targets and signaling pathways. By integrating reinforcement learning, active learning, and systems-level modeling, these methods can simultaneously modulate oncogenic networks, such as PI3K-BRD4 or PARP-associated DNA repair axes, addressing pathway redundancy and drug resistance in complex diseases. Emerging frameworks that combine knowledge graphs with large language models (LLMs) promise to enhance interpretability and scalability, enabling autonomous hypothesis generation and validation at scale. The focal graph concept, using centrality-based algorithms to extract meaningful hypotheses from high-dimensional data, further advances the evolution toward interpretable and explainable AI-driven discovery systems.

Nonetheless, several challenges must be resolved before these technologies can be fully realized in clinical settings. Issues such as algorithmic bias, model interpretability, a lack of transparency, and limited regulatory guidance remain key barriers to equitable and trustworthy AI deployment. Future regulatory frameworks must adapt to the dynamic, self-learning nature of AI systems, incorporating mechanisms for continuous validation, safety monitoring, and ethical oversight in clinical applications. By addressing these ethical, technical, and regulatory hurdles, AI-driven self-improving platforms can expedite the design of next-generation multi-target anticancer and other disease treatments, optimizing efficacy, minimizing resistance, and enabling personalized interventions aligned with the molecular complexity of cancer.

Looking ahead, the coming decade will witness the convergence of generative AI, multi-omics integration, and quantum-assisted computation in cancer drug discovery. Quantum machine learning and graph neural networks are expected to further improve molecular representation and property prediction at atomic precision. Additionally, the fusion of real-world clinical data, patient-derived organoid models, and experimental design driven by active learning will create continuously self-improving discovery ecosystems. These innovations will enable the shift from reactive cancer therapies to proactive, AI-guided precision medicine.

In summary, this review highlights the emergence of self-improving, AI-driven frameworks as a transformative paradigm for multi-target drug discovery. Through the integration of deep generative modeling, reinforcement, and active learning within the DMTL cycle, these approaches establish an autonomous and interpretable foundation for precision oncology and continuously evolving drug design. Continued interdisciplinary collaboration among computational scientists, molecular biologists, and clinicians will be pivotal in realizing the full translational potential of these next-generation intelligent drug design technologies.

## Figures and Tables

**Figure 1 ijms-26-11443-f001:**
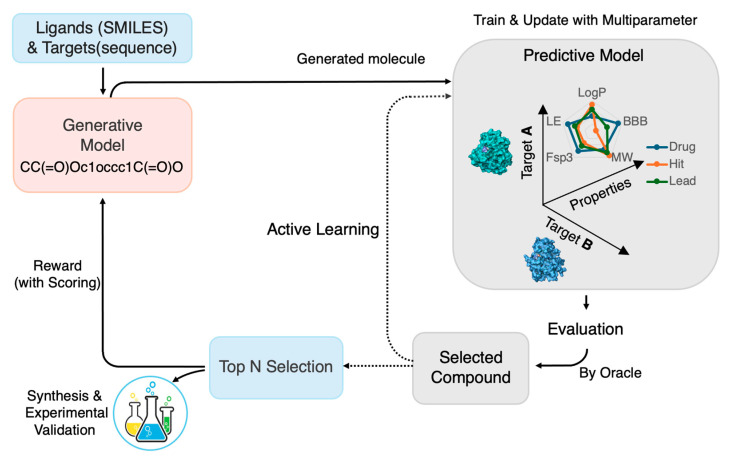
**Schematic flow of self-improving multi-target drug design via RL and AL.** This flow illustrates a closed-loop framework for de novo drug design that integrates generative modeling with RL and AL to optimize molecular candidates across multiple targets. The core DMTL workflow is denoted by solid arrows. Candidate molecules are initially generated and scored using in silico predictors for activity, drug-likeness, and toxicity. RL guides the subsequent generation process via a multi-objective reward function, informed by an oracle, a composite scoring system based on predictive models and/or experimental data. AL (dotted arrows) selects informative samples, based on uncertainty or diversity, to iteratively update the predictive models. This continuous feedback loop ensures the co-improvement of both the generative and predictive components, dramatically improving the efficiency of multi-target drug design.

**Figure 2 ijms-26-11443-f002:**
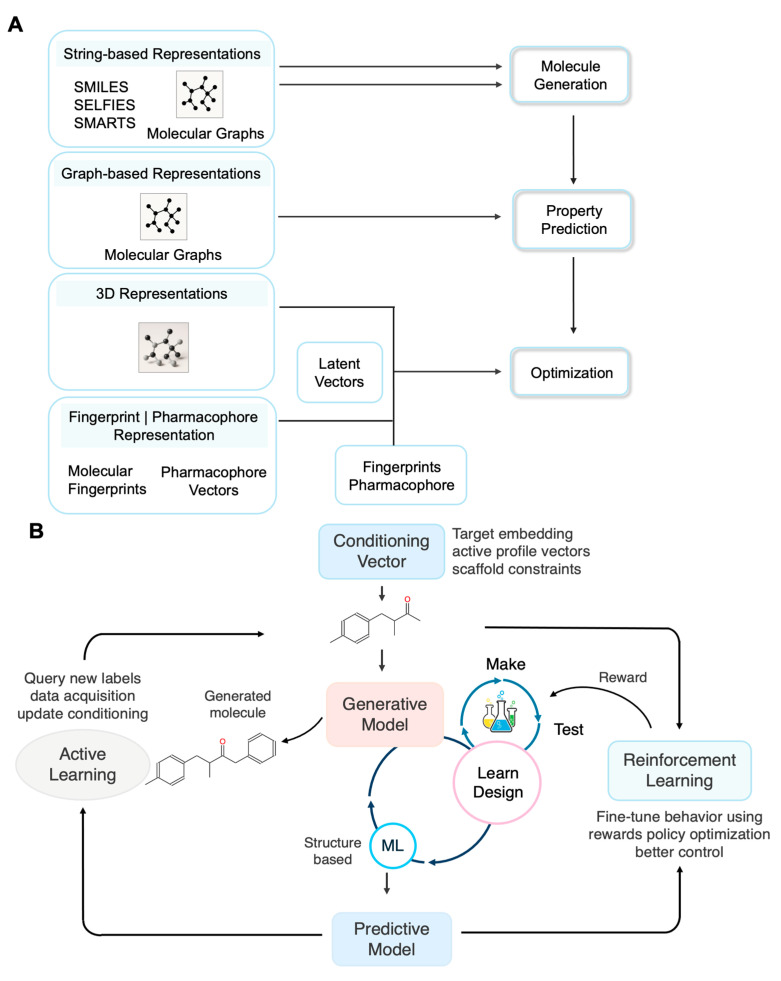
**Molecular Representations and Framework for Self-Improving Drug Design.** (**A**) Overview of molecular representation modalities used in DGMs for *de novo* drug design. Representations include string-based formats (e.g., SMILES, SELFIES), graph-based molecular graphs, 3D structural data, and pharmacophore or fingerprint vectors. These representations are encoded into latent vectors to support molecule generation, property prediction, and multi-objective optimization. (**B**) A closed-loop framework integrating generative models, predictive models, RL, and AL for self-improving molecular design. Conditioning vectors (e.g., target profiles, activity constraints) guide the generative model to propose candidate molecules. These molecules undergo structure-based testing, simulation, and evaluation via predictive models. RL further refines generative behavior using feedback (reward), while AL selects informative compounds for additional data acquisition and model retraining, accelerating the DMTL cycle.

**Figure 3 ijms-26-11443-f003:**
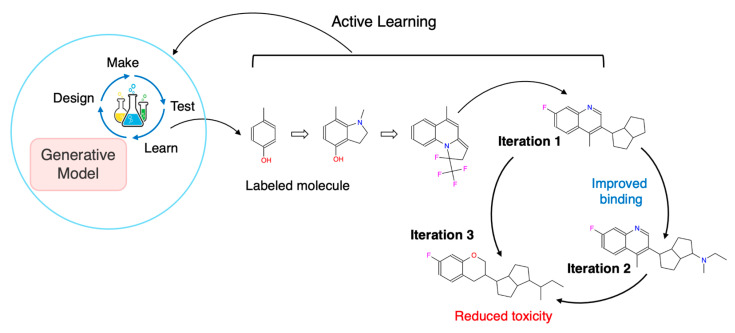
**Iterative AL Cycle for Multi-Target Inhibitor Design Using DGMs.** A closed-loop active learning strategy integrated with DGMs for the rational design of multi-target inhibitors. Beginning with a labeled molecule, the DMTL cycle is initiated to generate novel chemical structures. These candidates undergo experimental testing, and informative data points are queried to retrain the generative model. Through iterative refinement, molecules are optimized for improved polypharmacological profiles, which enhance binding to key targets and minimize off-target toxicity.

**Figure 4 ijms-26-11443-f004:**
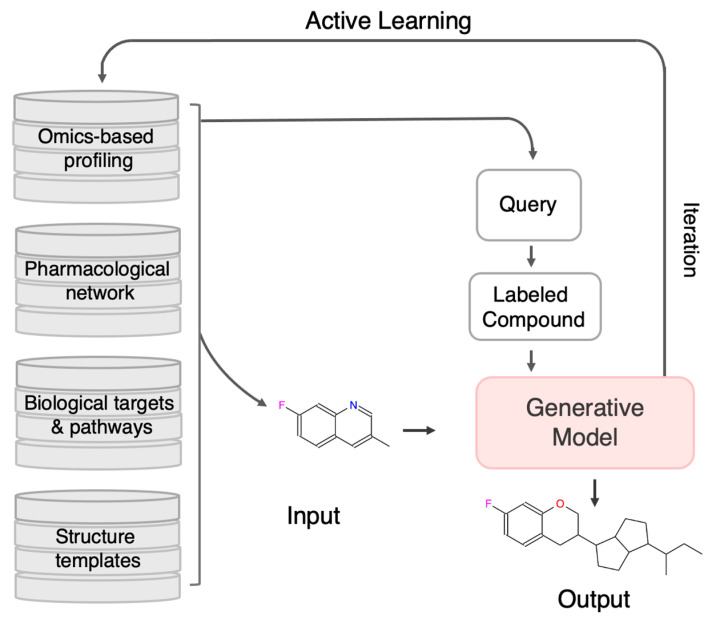
**Integrating Omics Data and Network Pharmacology for Multi-Target Cancer Drug Design via AL.** This framework showcases a systems-level approach to multi-target drug discovery in oncology, combining omics profiles (e.g., transcriptomics, genomics), pharmacological networks, cancer-related signaling pathways, and chemical structure templates. The generative model is conditioned using biological insights to design molecules with desired activity across multiple cancer-relevant targets. AL iteratively selects the most informative compounds for synthesis and testing, enabling continuous refinement of the model. By exploiting disease-specific networks such as those involving PI3K, BRD4, and other oncogenic drivers, this closed-loop process supports the rational design of polypharmacological agents tailored for complex cancer pathologies.

**Table 1 ijms-26-11443-t001:** The frameworks of protein-ligand interacting molecule generation.

Method	Protein Input	Ligand Input	Highlights	Usage
**AlphaFold 3**	Sequence	SMILES	End-to-end protein-ligand modeling; multimodal	Protein-ligand complex generation
**RFAA**	Sequence + ligand	Optional ligand	Ligand-aware folding; end-to-end structure refinement	Protein-ligand complex generation
**Pocket2Mol**	3D pocket (point cloud)	3D ligand	Direct 3D ligand generation from protein pocket	Target-aware ligand structure generation
**DiffSBDD**	3D pocket	3D ligand	Samples ligand-pocket pairs from learned 3D distributions	Target-aware ligand structure generation
**IDOLpro**	Sequence	SMILES	Inverse design for ligand generation targeting proteins; sequence-to-molecule generation	Multi-target-based ligand structure generation
**POLYGON**	2 protein targets (IDs)	SMILES	Polypharmacology-focused with dual-target optimization loop	Multi-target-based ligand structure generation
**CLM**	2 protein targets (ligand sets)	SMILES	Fine-tuning with dual ligand templates; pharmacophore fusion	Multi-target-based ligand structure generation

## Data Availability

All data are available in the main text.

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
