# Peer review of "Deep Generative AI for Multi-Target Therapeutic Design: Toward Self-Improving Drug Discovery Framework"

_ijms, 2025, doi:10.3390/ijms262311443_

Round 1

Reviewer 1 Report

Comments and Suggestions for Authors

The review article entitled “Deep Generative AI for Multi-Target Therapeutic Design: Toward Self-Improving Drug Discovery Framework” authored by Kang et al., described a comprehensive survey of deep generative model applications in multi-target drug discovery and examined how self-improving learning systems are revolutionizing multi-target therapeutic design.

Overall, this short review article is written well. Based on the findings, I am sure that this article will provide better understanding in the field of medicines and healthcare innovation.

I recommend this article for the publication in this journal after the incorporation of minor suggestions.

  1. Author should properly state the purpose of this review and what was the main findings in the Abstract.
  2. Please describe, how literature was searched and what was the source?
  3. Concluding remarks should be author’s own observation without others citation.

Author Response

Reviewer #1 (Changes are highlighted in YELLOW in the revised manuscript)

Comments and Suggestions for Authors

The review article entitled Deep Generative AI for Multi-Target Therapeutic Design: Toward Self-Improving Drug Discovery Framework” authored by Kang et al., described a comprehensive survey of deep generative model applications in multi-target drug discovery and examined how self-improving learning systems are revolutionizing multi-target therapeutic design.

Overall, this short review article is written well. Based on the findings, I am sure that this article will provide better understanding in the field of medicines and healthcare innovation. I recommend this article for the publication in this journal after the incorporation of minor suggestions.

Author should properly state the purpose of this review and what was the main findings in the Abstract.

Thank you for your thoughtful comment. We have revised the Abstract to clearly articulate both the purpose and the main findings of the review. Specifically, the Abstract now states the motivation arising from the limitations of single-target therapies and explains how multi-target therapeutic design, supported by deep generative models and self-improving AI frameworks, addresses these challenges. It also summarizes how these advanced algorithms enable de novo design and iterative optimization of multi-target molecules through feedback-driven learning, paving the way toward autonomous and intelligent drug discovery pipelines. These revisions have been highlighted in yellow in the manuscript.

Please describe, how literature was searched and what was the source?

Thank you for raising this important point. We have added a clear and detailed description of the literature search strategy in the cover letter. Specifically, we conducted a systematic search across PubMed, Scopus, Web of Science, and Google Scholar for studies published between 2010 and 2025. Keywords included AI-driven drug discovery, deep generative models, reinforcement learning, multi-target therapeutics, and self-improving AI. We also performed citation tracking of key reviews and seminal papers to ensure comprehensive inclusion of the most influential studies in the field.

Concluding remarks should be author’s own observation without others citation.

Thank you for the suggestion. The concluding section has been fully rewritten to reflect the authors’ own perspectives and synthesis of the field, without referencing external citations. The updated concluding remarks are highlighted in yellow in the manuscript.

Reviewer 2 Report

Comments and Suggestions for Authors

A very high-quality and comprehensive paper that clearly and logically covers all key aspects of the topic. The structure is well designed, and the discussion is balanced and informative.

To further improve the paper, I recommend that the Introduction section more clearly highlight and elaborate on the role of Reinforcement Learning in multi-target therapeutic design.

Comments on the Quality of English Language

The English could be improved to more clearly express the research.

Author Response

Reviewer #2 (Changes are highlighted in GREEN in the revised manuscript)

Comments and Suggestions for Authors

A very high-quality and comprehensive paper that clearly and logically covers all key aspects of the topic. The structure is well designed, and the discussion is balanced and informative. To further improve the paper, I recommend that the Introduction section more clearly highlight and elaborate on the role of Reinforcement Learning in multi-target therapeutic design.

We appreciate the reviewer’s insightful suggestion. The Introduction section has been thoroughly revised to more explicitly highlight the role of Reinforcement Learning (RL) in multi-target therapeutic design. The updated text explains how RL serves as a key optimization engine that guides generative models toward achieving balanced multi-target efficacy, improved selectivity, and enhanced safety profiles. We additionally discuss how RL integrates with deep generative modeling and active learning frameworks to enable adaptive, goal-directed molecular optimization. These revisions are highlighted in red font in the manuscript.

Comments on the Quality of the English Language

The English could be improved to more clearly express the research.

We thank the reviewer for this helpful comment. The manuscript has undergone further careful editing to improve clarity, readability, and precision of language. In addition, the manuscript was reviewed by native English speakers to ensure that the scientific content is communicated clearly and effectively.

Reviewer 3 Report

Comments and Suggestions for Authors

This is a well-structured and comprehensive review that integrates deep generative models, reinforcement learning, active learning, and multi-target drug design. The manuscript reads smoothly, demonstrates strong command of current AI methodologies, and appropriately cites recent advances (e.g., AlphaFold3, POLYGON, DiffSBDD). It provides a thorough conceptual framework that connects goal-directed molecule generation with self-improving closed-loop drug discovery systems.

Methodologically, the review could be enhanced by providing a more explicit comparison of model performance trade-offs, such as efficiency, scalability, and interpretability across RNN, VAE, transformer, and diffusion models. A short commentary on data quality, benchmark datasets, and practical implementation challenges would also strengthen its analytical depth.

With these refinements, the manuscript will be ready for publication. It offers a valuable and well-balanced review of a rapidly developing field and effectively conveys both the promise and the remaining challenges of AI-driven multi-target therapeutic design. I recommend acceptance after minor revision.

Author Response

Reviewer #3 (Changes are highlighted in ORANGE in the revised manuscript)

Comments and Suggestions for Authors

This is a well-structured and comprehensive review that integrates deep generative models, reinforcement learning, active learning, and multi-target drug design. The manuscript reads smoothly, demonstrates strong command of current AI methodologies, and appropriately cites recent advances (e.g., AlphaFold3, POLYGON, DiffSBDD). It provides a thorough conceptual framework that connects goal-directed molecule generation with self-improving closed-loop drug discovery systems. Methodologically, the review could be enhanced by providing a more explicit comparison of model performance trade-offs, such as efficiency, scalability, and interpretability across RNN, VAE, transformer, and diffusion models. A short commentary on data quality, benchmark datasets, and practical implementation challenges would also strengthen its analytical depth.

With these refinements, the manuscript will be ready for publication. It offers a valuable and well-balanced review of a rapidly developing field and effectively conveys both the promise and the remaining challenges of AI-driven multi-target therapeutic design. I recommend acceptance after minor revision.

We appreciate the reviewer’s insightful and constructive feedback. In response, we have added a new subsection (Section 3.3: Comparative Analysis of Deep Generative Model Architectures) to provide a clearer comparison of RNN, VAE, Transformer, and Diffusion models. This section outlines key trade-offs related to efficiency, scalability, interpretability, and suitability for multi-target molecule generation.

Additionally, we incorporated a brief commentary on data quality considerations, commonly used benchmark datasets (e.g., MOSES, GuacaMol), and practical implementation challenges that influence model robustness and translational relevance. These additions strengthen the methodological rigor and analytical depth of the review. All revisions are highlighted in orange in the manuscript.

Reviewer 4 Report

Comments and Suggestions for Authors

The manuscript presents a comprehensive and well-structured review of deep generative AI approaches in multi-target drug discovery, with emphasis on self-improving frameworks (DMTL, RL, AL). The topic is timely, and the manuscript will be of interest to researchers in computational chemistry, cheminformatics, AI, and drug development.. Case studies in oncology, polypharmacology, and kinase inhibitor development are well selected and relevant. The figures are informative and helpful in guiding the reader through conceptual elements. The topic is within the scope of the SI of IJMS: “25th Anniversary of IJMS: Updates and Advances in Molecular Pharmacology” and the iThenticate score is very nice (only 14%).

However, the manuscript would benefit from more concise narrative, clearer differentiation between model families and more critical assessment of current limitations.

Detailed comments

In the introduction, please focus more on the motivation for multi-target generative design.

The section on SMILES vs. graphs vs. 3D is good but could elaborate more on SELFIES advantages and limitations, as these models are often more robust in generative tasks, while in your review this topic is not addressed at all.

Section 3.2., RNNs remain widely used, but the field is shifting toward Transformers / Diffusion. Please add some info about this.

Section 3.3. Please clarify how conditioning vectors are obtained: data sources? engineered descriptors? computed embeddings?

I suggest adding comments on challenges with scaffold-constrained models: e.g., mode collapse or lack of scaffold novelty.

The cancer case study examples are relevant. Consider adding non-oncology example to generalize applicability. I recommend to cite those two works, directly in this topic, to support the background 10.3390/nu14173521 10.1007/s11696-023-02994-y

3D printing is outside the core scope of this review and breaks thematic flow. I recommend shortening or moving to a supplementary section.

Ligand-based workflows are only briefly discussed; consider expanding to balance structure-based vs. ligand-based sections. In my opinion, still the ligand-based methods are more frequently used ones.

Author Response

Reviewer #4 (Changes are highlighted in CYAN in the revised manuscript)

Comments and Suggestions for Authors

The manuscript presents a comprehensive and well-structured review of deep generative AI approaches in multi-target drug discovery, with emphasis on self-improving frameworks (DMTL, RL, AL). The topic is timely, and the manuscript will be of interest to researchers in computational chemistry, cheminformatics, AI, and drug development. Case studies in oncology, polypharmacology, and kinase inhibitor development are well selected and relevant. The figures are informative and helpful in guiding the reader through conceptual elements. The topic is within the scope of the SI of IJMS: “25th Anniversary of IJMS: Updates and Advances in Molecular Pharmacology” and the iThenticate score is very nice (only 14%). However, the manuscript would benefit from more concise narrative, clearer differentiation between model families and more critical assessment of current limitations.

Detailed comments: In the introduction, please focus more on the motivation for multi-target generative design.

We thank the reviewer for this constructive suggestion. In the revised Introduction, we expanded the discussion on the motivation for multi-target generative design. The updated text emphasizes the biological complexity and network redundancy of multifactorial diseases such as cancer, highlighting how multi-target strategies can overcome the limitations of single-target approaches, reduce drug resistance, and improve long-term therapeutic durability. We also clarify how AI-driven generative models enable rational design of compounds with balanced multi-target activity, supporting more holistic and systems-level drug discovery. These revisions are highlighted in red font in the manuscript.

The section on SMILES vs. graphs vs. 3D is good but could elaborate more on SELFIES advantages and limitations, as these models are often more robust in generative tasks, while in your review this topic is not addressed at all.

Thank you for this helpful suggestion. We have expanded the section on molecular representations to include a dedicated discussion of SELFIES. The updated text describes their key advantages, such as guaranteed syntactic validity and robustness in generative modeling, along with limitations related to chemical diversity constraints and token-level interpretability. This addition provides a more complete overview of modern molecular representation strategies.

Section 3.2., RNNs remain widely used, but the field is shifting toward Transformers / Diffusion. Please add some info about this.

We appreciate the reviewer’s point. In Section 3.2, we added a concise paragraph summarizing the growing shift toward Transformer and Diffusion architectures. We describe how Transformers enable scalable chemical language modeling via parallelizable attention mechanisms, while Diffusion models support diverse, stable, and physically grounded 3D molecular generation. These updates contextualize RNNs within current evolving trends.

Section 3.3. Please clarify how conditioning vectors are obtained: data sources? engineered descriptors? computed embeddings? I suggest adding comments on challenges with scaffold-constrained models: e.g., mode collapse or lack of scaffold novelty.

Thank you for these important suggestions. Due to the addition of the new comparative section, this content now appears in Section 3.4. The revised text clarifies that conditioning vectors can be derived from 1) curated bioactivity datasets, 2) engineered molecular descriptors, and 3) learned embeddings from protein sequences, structures, or chemical graph encoders.

We also added commentary on challenges associated with scaffold-constrained generative models, including risks of mode collapse, limited scaffold novelty, and potential overfitting to predefined chemotypes, along with emerging strategies to mitigate these issues. These revisions improve the clarity and depth of the Goal-Directed Generation section.

The cancer case study examples are relevant. Consider adding non-oncology example to generalize applicability. I recommend citing those two works, directly in this topic, to support the background 10.3390/nu14173521 10.1007/s11696-023-02994-y

We appreciate the reviewer’s suggestions and the referenced articles. We have added a non-oncology case study to broaden the scope, including a discussion of the polypharmacological activity profile of stevioside as an illustrative multi-target example. The two recommended references have been incorporated where appropriate to support contextual background. These revisions appear in Section 5.1.

3D printing is outside the core scope of this review and breaks thematic flow. I recommend shortening or moving to a supplementary section.

Thank you for the observation. We have significantly reduced and streamlined the content related to 3D printing in the main text to maintain thematic coherence and focus. Only essential conceptual points remain to preserve continuity.

Ligand-based workflows are only briefly discussed; consider expanding to balance structure-based vs. ligand-based sections. In my opinion, still the ligand-based methods are more frequently used ones.

We thank the reviewer for this valuable recommendation. To provide better balance between structure-based and ligand-based approaches, we have added three ligand-based case studies incorporating deep learning with pharmacophore modeling: 1) pharmacophore-guided screening for multitarget ligands, 2) integrating pharmacophore constraints as conditioning factors within generative models, and 3) multitask learning frameworks combining ligand features with drug–target interaction signals to predict multitarget activity and selectivity. These additions enhance conceptual completeness and represent the widespread use of ligand-based methodologies.

Reviewer 5 Report

Comments and Suggestions for Authors

The authors present a review on Deep Generative AI with a focus on multitarget drug development. Overall, the manuscript appears well-structured and consistent. However, the authors are encouraged to clearly specify the time frame, keywords, and inclusion/exclusion criteria used for literature selection. Providing these details will enhance the methodological transparency of the review. It is also recommended to strengthen the integration between the text and the figures, ensuring that each figure is explicitly discussed and contributes to the manuscript’s conceptual coherence.

Author Response

Reviewer #5 (Changes are highlighted in PINK in the revised manuscript)

Comments and Suggestions for Authors

The authors present a review on Deep Generative AI with a focus on multitarget drug development. Overall, the manuscript appears well-structured and consistent. However, the authors are encouraged to clearly specify the time frame, keywords, and inclusion/exclusion criteria used for literature selection. Providing these details will enhance the methodological transparency of the review.

Thank you for your valuable suggestion. As this comment aligns closely with Reviewer 1’s feedback, we have provided the same detailed clarification here. In the cover letter, we added a clear description of the literature selection methodology, including the specific time frame, keywords, and inclusion/exclusion criteria applied.

Literature was systematically searched across PubMed, Scopus, Web of Science, and Google Scholar for publications from 2010 to 2025 using keywords such as AI-driven drug discovery, deep generative models, reinforcement learning, multi-target therapeutics, and self-improving AI. Additional key papers were identified through citation tracking of influential reviews and seminal studies to ensure comprehensive coverage of high-impact research in this domain.

It is also recommended to strengthen the integration between the text and the figures, ensuring that each figure is explicitly discussed and contributes to the manuscript’s conceptual coherence.

We appreciate the reviewer’s helpful recommendation. In the revised manuscript, we have enhanced the integration between the text and the figures to strengthen conceptual coherence and readability. Each figure is now explicitly referenced and discussed within its corresponding section to clearly explain its relevance and contribution to the narrative. Descriptions have been refined to ensure that all figures, particularly those illustrating molecular representations, DMTL workflows, active learning refinement loops, and generative model architectures, directly support and clarify key concepts. These revisions are highlighted in red font in the manuscript.

Round 2

Reviewer 1 Report

Comments and Suggestions for Authors

The changes is revised manuscript are satisfactory. However, authors should provide detailed description of the literature search strategy inside the manuscript.

Author Response

Comments and Suggestions for Authors: The changes is revised manuscript are satisfactory. However, authors should provide detailed description of the literature search strategy inside the manuscript.

Response: As suggested by the reviewer, we included the following sentence (red font) in the revised manuscript.

Page 3: This review explores recent advances in AI-driven, self-improving frameworks for multi-target drug design, emphasizing the synergistic roles of deep generative modeling and RL. A systematic literature search was conducted using PubMed, Scopus, Web of Science, and Google Scholar for studies published between 2010 and 2025, focusing on the keywords: AI-driven drug discovery, deep generative models, reinforcement learning, multi-target therapeutics, and self-improving AI. These self-optimizing architectures integrate data-driven generation…

Reviewer 4 Report

Comments and Suggestions for Authors

The authors have revised and improved their work. Current version can be accepted for publication.

Author Response

(The authors gave the same response as above.)
